# 4,5-Bis(arylethynyl)-1,2,3-triazoles—A New Class of Fluorescent Labels: Synthesis and Applications

**DOI:** 10.3390/molecules27103191

**Published:** 2022-05-17

**Authors:** Anastasia I. Govdi, Polina V. Tokareva, Andrey M. Rumyantsev, Maxim S. Panov, Johannes Stellmacher, Ulrike Alexiev, Natalia A. Danilkina, Irina A. Balova

**Affiliations:** 1Institute of Chemistry, Saint Petersburg State University (SPbU), Universitetskaya nab. 7/9, 199034 Saint Petersburg, Russia; tokareva.spbu@gmail.com (P.V.T.); m.s.panov@spbu.ru (M.S.P.); danilkina.natalia@gmail.com (N.A.D.); 2Department of Genetics and Biotechnology, Saint Petersburg State University (SPbU), Universitetskaya nab. 7/9, 199034 Saint Petersburg, Russia; rumyantsev-am@mail.ru; 3Department of Physics, Institute of Experimental Physics, Freie Universität Berlin, Arnimalllee 14, 14195 Berlin, Germany; jcart@zedat.fu-berlin.de (J.S.); ulrike.alexiev@fu-berlin.de (U.A.)

**Keywords:** 1,2,3-triazoles, 1,3-diynes, azide–alkyne cycloaddition, Sonogashira cross-coupling, fluorescence, bioimaging, cytotoxicity

## Abstract

Cu-catalyzed 1,3-dipolar cycloaddition of ethyl 2-azidoacetate to iodobuta-1,3-diynes and subsequent Sonogashira cross-coupling were used to synthesize a large series of new triazole-based push–pull chromophores: 4,5-bis(arylethynyl)-1*H*-1,2,3-triazoles. The study of their optical properties revealed that all molecules have fluorescence properties, the Stokes shift values of which exceed 150 nm. The fluorescent properties of triazoles are easily adjustable depending on the nature of the substituents attached to aryl rings of the arylethynyl moieties at the C4 and C5 atoms of the triazole core. The possibility of 4,5-bis(arylethynyl)-1,2,3-triazoles’ application for labeling was demonstrated using proteins and the HEK293 cell line. The results of an MTT test on two distinct cell lines, HEK293 and HeLa, revealed the low cytotoxicity of 4,5-bis(arylethynyl)triazoles, which makes them promising fluorescent tags for labeling and tracking biomolecules.

## 1. Introduction

Fluorescent technologies are known as one of the most rapidly developing areas of biomedical research. Fluorescent imaging plays a crucial role in cell labeling, enzyme activity measurements, and tumor diagnosis and therapy, including real-time tumor assessment during surgery [1]. In particular, fluorescence imaging and flow cytometry are considered as reliable and highly sensitive methods for determining the localization of pathogens in the human body [2,3,4]. The choice of a fluorescent probe with appropriate properties that allows identification and quantitative determination of biotargets in vivo using a non-destructive and non-invasive analysis is of crucial importance [5,6,7,8]. Therefore, the search for new chemical compounds as a basis for the development of useful fluorescent labels is considered an urgent task for chemists and biologists.

The nature of chemical structure in the search for new fluorescent compounds is of crucial importance. These molecules must have the necessary photophysical activity, along with the ability to be vectorized by reactive functional groups to bind to a biotarget under physiological conditions [9]. In addition, the fluorescent dyes must meet requirements, such as ease of synthesis and chemical modification, stability during preparation, storage and use, high photostability, and low toxicity [10,11].

Despite the fact that a variety of fluorophores is known [12], in the last decade, chemists and biologists have been interested in new fluorescent compounds based on 1,2,3-triazole [13]. Triazole-based fluorophores can be easily synthesized using a “click chemistry” approach, i.e., Cu-catalyzed azide–alkyne cycloaddition (CuAAC) [14,15]. Moreover, the fluorescent properties of triazoles are strongly dependent on the nature of the substituents attached to the N1, C4, and C5 positions, which can be varied easily through CuAAC. Triazole-fluorescent dyes are of great interest in the field of environmental-sensitives probes [16] and fluorogenic compounds [17].

Known examples of triazole-based fluorophores have a triazole ring connected to both a donor-type group (EDG, D) and an acceptor-type moiety (EWG, A) (Figure 1). However, the role of triazole as a π-spacer between the electron-donating and electron-withdrawing parts is ambiguous [18,19,20,21,22,23]. Thus, the triazole ring includes a lone pair of nitrogens in the conjugation system in the case when the EWG/EDG are attached to the N1/N2 atom and to the C4/C5 atom (Figure 1A–E), or it serves as a π-spacer in the case of C4/C5 EWG/EDG-disubstituted triazoles (Figure 1F). An interesting example of mixed-substitution-type bis(triazolyl)conjugated systems with restricted rotation of the aryl fragment at the C5 position of the triazole has also been reported (Figure 1D). All of these structural changes strongly influence the fluorescent properties of triazoles [13].

Recently, we reported the synthesis of 5-aryl-4-arylethynyltriazoles with a 1,2,3-triazole core as a π-spacer between EWG and EDG, attached to the C4 and C5 positions. 5-Aryl-4-arylethynyltriazoles were found to be fluorescent compounds with high Stokes shifts (>100 nm) and promising fluorescence quantum yields (15–64%) [23]. In order to study whether the extension of the conjugated π-system would improve the fluorescent properties of the triazole-based fluorophores, we decided to replace the aryl ring at the C5 atom with an arylethynyl moiety.

Therefore, we report the efficient synthetic rout towards 4,5-bis(arylethynyl)-1,2,3-triazole bearing EWG and EDG at *ortho-* and *para*-positions of aryl rings here. We have demonstrated that easily synthetically accessible 4,5-bis(arylethynyl)-1,2,3-triazoles are fluorescent in a wider spectral range (350–600 nm). Moreover, some derivatives of new 4,5-bis(arylethynyl)-1,2,3-triazoles have extremely high Stokes shift values (up to 230 nm). The photophysical properties of 4,5-bis(arylethynyl)-1,2,3-triazoles are strongly dependent on the relative orientations of the EWG and EDG at the C4 and C5 positions and on the type of the substitution (either *ortho*- or *para*-). To reach the highest fluorescence QY, it is important to have an EDG*-ortho*-substituted arylethynyl ring at the C5 position, along with an EWG*-para*-substituted arylethynyl ring at the C4 atom. For the highest redshift of emission, the orientation of the groups must be inverted.

## 2. Results and Discussion

### 2.1. Synthesis of 4,5-Bis(arylethynyl)triazoles 

The target 4,5-bis(arylethynyl)-1,2,3-triazoles **5** with electron-donating and electron-withdrawing groups in arylethynyl moieties at the C4 and C5 atoms of the triazole ring, were obtained using CuAAC of 1-iodobuta-1,3-diynes **1a**–**d** with 2-azidoethylacetate **2,** followed by the Sonogashira coupling of 4-ethynyl-5-iodo-1,2,3-triazoles **3** with arylacetylenes **4a**–**f** with Pd(PPh_3_)_4_/K_3_PO_4_ as a catalytic system (Figure 1). This approach opens access to the wide range of unsymmetrically substituted 4,5-bis(arylethynyl)-1,2,3-triazoles by varying the substituents in the starting iododiacetylene and arylalkyne. The choice of 2-azidoethylacetate as a dipole is based on the potential ability of the COOEt group for further derivatization. Both steps, i.e., cycloaddition and the Sonogashira coupling, were carried out under optimized conditions [24]. The reactions proceeded without any difficulties and gave 4,5-bis(arylethynyl)triazoles **5a**–**m** in moderate to high yields.

The X-ray data obtained for triazole **5d** revealed the non-planar geometry of this compound. Thus, the C4-aryl ring lies in the plane of the diethynyltriazlole system. On the contrary, the aryl ring at the C5 position and the triazlole ring are almost orthogonal. The dihedral angle between the 4-*N,N*-dimethylaminophenyl and the triazole rings was found to be 79°.

Then, we turned to the investigation of the photophysical properties of the synthesized triazoles.

### 2.2. Investigation of Optical Properties 

The UV−Vis absorption and photoluminescence (PL) spectra of the synthesized series of compounds **5a**–**m** were obtained for tetrahydrofuran (THF) solutions of triazoles. The absorption maxima in the absorption spectra of the synthesized compounds were observed in the region between 250 and 450 nm. The absorption spectra differed depending on the nature of the substituents at the C4 and C5 atoms (Figure 2).

The THF solutions of all of the synthesized triazoles display fluorescent properties (Figure 3, Table 1). The compounds **5a**–**d,j,h,i** with a *N,N*-dimethylaminophenyl group exhibited significant bathochromic shifts (447–609 nm) compared to the triazoles bearing MeOPh moiety. The strongest bathochromic shift of the emission maximum was observed in the case of triazoles **5h** (λ ex/em = 302/609 nm) and **5i** (λ ex/em = 299/588 nm). However, the fluorescence intensity of these compounds was much lower than for other triazoles **5**. Stokes shifts of **5h** and **5i** were 16.692 and 16.438 cm^−1^, respectively, whereas their absolute quantum yields reached only 5%. 

The push–pull triazole-based dyes bearing aromatic substituents with EDGs (NMe_2_) in the C5 positions and EWGs (CN, NO_2_, Cl) in the C4 positions demonstrated a blue-green fluorescence at around of 447–514 nm (Figure 3, left) and high Stokes shifts (Table 1).

Interestingly, the dye **5k** with the strongest EWG (NO_2_) and the strongest EDG (NMe_2_) had the strongest blueshift of the emission maximum (λ ex/em = 272/398 nm) compared to triazoles **5a**–**d,j**, despite the fact that, typically, the presence of a nitro group in the chromophore leads to red-shifts of the emission maximum. In addition, this compound had the lowest emission intensity among all of the prepared derivatives. A similar effect was observed by Zhu and co-authors for triazole containing a nitrophenyl substituent at the nitrogen atom N1 [25].

Thus, triazoles **5a** and **5c** bearing *para*-CN group at the C4 arylethynyl unit, along with an *ortho/para* Me_2_N group at the C5 arylethynyl moiety, can be proposed as the most promising compounds for further studies. Both triazoles exhibited high fluorescence quantum yields (Φ_f_ = 60% (**5a**), Φ_f_ = 22% (**5c**)) with the emission maxima lying in the blue (492 nm) and green (514 nm) regions, respectively. It is worth noting that compound **5a** had the largest Stokes shift of 18.434 cm^−1^ (234 nm) in comparison with other synthesized triazoles.

Triazoles **5e**–**g** containing methoxyphenylethynyl substituents showed violet fluorescence at ~395 nm; the absolute fluorescence quantum yield of compounds **5e** and **5f** reached 40% and 56%, respectively. At the same time, the fluorescence quantum yield of triazole **5g** dropped to 7%.

The position of the substituents in the phenylethynyl groups (*ortho*- or *para-*) is extremely important. Comparing pairs of triazoles with similar substituents that differ only in the *ortho/para* position of the substituents, i.e., pair 1 (C4: *para* CN and C5: *ortho*-NMe_2_ (**5a**)/*para*-NMe_2_ (**5c**)); pair 2 (C4: *para* Cl and C5: *ortho*-NMe_2_ (**5b**)/*para*-NMe_2_ (**5d**)); and pair 3 (C4: *para* Cl and C5: *ortho*-OMe (**5f**)/*para*-OMe (**5g**)), it is obvious that QY values are always higher for the *ortho*-derivatives. Moreover, the Stokes shift values for all pairs have a similar trend: the values are smaller for *para*-isomers (Figure 4, Table 1).

The first results allowed us to choose five compounds **5b**–**e,i** for further studies in aqueous media suitable for further biological experiments (water, phosphate buffered (PBS, pH 7.4), and synthetic cell culture medium (DMEM)). All solutions were prepared from stock solutions of triazoles in DMSO. The final concentration of DMSO did not exceed 2%, *v*/*v*. All absorption maxima lay in the region of 250–450 nm (Figure 5). It is important to note that in the emission spectra of all triazoles in aqueous media, a redshift of emission was observed, regardless of the nature of the substituent at the C5 position of the triazole core. This property is crucial for the use of triazole-based dyes under physiological conditions. However, a decrease in fluorescence intensity was observed.

The absorption and fluorescence spectra of triazoles in aqueous media depend on the nature of the aqueous solution (Figure 5). The fluorescence for the solutions of most compounds in PBS buffer and DMEM is more intense in comparison with those observed for the solutions in water (Figure 5). The results obtained indicate that the fluorescence intensity of 4,5-bis(arylethynyl)-1,2,3-triazoles is affected by both the pH of the medium and the presence of various organic compounds (amino acids, vitamins, and proteins) presented in DMEM.

The emission maxima (λ_Em_), Stokes shift ($\bar{v}$_Abs_$-\bar{v}$_Em_)/cm^−1^, quantum yield (Φ_f_), and lifetimes (τ, ns) of triazoles **5a**–**m** in THF and water are listed in Table 1.

It should be noted that the expansion of the conjugated system does not introduce fundamental changes in the photophysical properties of 4,5-bis(arylethynyl)-1,2,3-triazoles compared to 5-aryl-4-arylethynyltriazoles [23]. However, it allows shifting the excitation wavelength to the red region of the spectrum by 30–50 nm and obtaining more examples of fluorescent triazoles for further selection of the optimal compounds.

### 2.3. Modification of ***5b***,***h*** by ”Clickable“ Groups, Study of Their Optical Properties, and Application as Fluorescent Dyes

With different fluorescent triazoles in hand, we turned to converting the ester group into various functional groups suitable for further conjugation with biomolecules. A total of 2 triazoles **5b**,**h** with high fluorescence intensity, suitable emission wavelengths (λ_em_ > 440 nm), and large Stokes shifts were selected. The ester group attached to the triazole core through the N1 atom can be considered as a universal functional group for the further conversion into various chemical handles for the conjugation with biological targets.

The triazoles containing amino-reactive groups, such as isothiocyanate (NCS), can be used for the protein labeling. In addition, the functionalization of triazoles with an azido group allows using an azide–alkyne cycloaddition to introduce triazole-based fluorescent labels into biomolecules premodified by alkyne-type functional groups.

To synthesize triazole modified with N_3_ and NCS groups, esters **5b**,**h** were first hydrolyzed to the corresponding acids **6a**,**b** (Figure 2). The obtained acids **6a**,**b** were clean enough for use in further steps without additional purification. In addition, purification was complicated by the poor solubility of acids in the various organic solvents.

The reaction of acids **6a**,**b** with 3-azidopropane-1-amine **7** as a crosslinking spacer in the presence of the coupling reagent (HATU/DIPEA) gave the corresponding amides with the azide group **8a**,**b** in good yields. The azides **8a**,**b** are not only appropriate substrates for “click” reactions with biological objects, but they can also be used as precursors to produce compounds with an isothiocyanate group. Thus, isothiocyanate **9a** was synthetized from azide **8a**, using the Staudinger reaction with triphenylphosphine and carbon disulfide.

First, the UV–Vis absorption spectra and emission spectra were obtained for all target molecules **8a,b,** and **9a** in water/DMSO, PBS/DMSO, and DMEM/DMSO mixtures. (The concentration of DMSO did not exceed 2%, *v*/*v*.) (See Figure 6.) The absorption and fluorescence spectra of the studied compounds changed significantly depending on the solvent used (Figure 6). 

Only small visible changes in the UV spectra of azides **8a**,**b,** and isothiocyanate **9a** in water and DMEM were observed. However, a small (6–10 nm) hypsochromic shift was observed in the case of the water, PBS, and DMEM for **8b** (Figure 6). 

A small solvatochromic effect could be observed in PBS (~6.0 nm) for **9a**, whereas a significant bathochromic shift was demonstrated by azide **8a** (257→333 nm).

The fluorescence intensity of the solutions of the most compounds in PBS buffer and DMEM increased in comparison with the fluorescence intensity for the corresponding solutions in water (Figure 6). 

The emission intensity for **8a** and **9a** in the mixture of water/DMSO was almost the same, whereas in the phosphate buffer, the fluorescence intensity of isothiocyanate **9a** decreased dramatically. However, in the culture medium DMEM, the intensity of the fluorescence of **9a** increased. Moreover, the emission peaks of **9a** exhibited a solvatochromic blueshift at ~465 nm (Figure 6F).

Azide **8b** was almost non-fluorescent in water although the fluorescence intensity increased slightly in PBS and DMEM. In addition, there was a shift of the maximum fluorescence from 530 nm in PBS to 550 nm in DMEM (Figure 6D–F).

The results obtained indicate that the fluorescence intensity of 4,5-diethynyl-1*H*-1,2,3-triazoles is affected by both pH value and by the presence of various organic compounds (amino acids, vitamins, and proteins) presented in DMEM.

All of the data related to the optical properties of THF and water solutions of **8a**,**b** and **9a** in are given in Table 2.

Next, we turned to the labeling of proteins with synthesized triazole-based fluorescent reagents. Fluorescence-based assays are widely used in molecular biology in a vast variety of applications, such as the investigation of molecular interactions; the measurement of enzymatic activities; and the study of signal transduction and distribution of molecules, organelles, or cells [26]. Most of these methods require labelling of the protein of interest with one or more fluorophores [27]. Thus, amine-reactive fluorescent dyes, such as fluorescein isothiocyanate (FITC), label biomolecules by forming a covalent bond with the formation of the substituted thioureas [28]. 

Covalent binding can be provided in a variety of ways, depending on the structure of the object under study. To demonstrate the possibility of the protein labelling by the synthesized dyes (**9a**, **8a**) via covalent binding, we used two binding methods: the azide–alkyne cycloaddition (for **8a**) and the interaction of the isothiocyanate group with the primary amino groups of proteins (for **9a**). 

In the current study, different proteins were labeled with **9a** using a modification through the fast and simple protocols that have been developed for FITC [29]. Aldolase and bovine serum albumin (BSA), that is often used for the demonstration of protein labeling [29], were labelled separately (Figure 7a). Then, the labeling of a protein standard consisting of five different proteins (phosphorylase b, BSA, ovalbumin, carbonic anhydrase, and recombinant KNOX-HD protein) was performed in a single reaction mixture (Figure 7b). The choice of proteins in the standard mixture was mainly based on their different molecular weights, which allows for their efficient separation using gel electrophoresis. We found that phosphorylase b, bovine serum albumin (BSA) could be efficiently labelled with **9a**, while the labelling of ovalbumin, carbonic anhydrase, and recombinant protein KNOX3-HD was less efficient. Isothiocyanate groups interact not only with the amino terminus of the protein, but also with the side chain of the lysine residues on the surface of the protein. The observed selectivity of labeling might be due to the different amounts of lysine residues on the surface of the proteins in the standard mixture (Figure 7).

It was shown that the binding of isothiocyanate **9a** was most effective with proteins, with a mass that is greater than 66 kDa.

In addition, we carried out the labeling of bovine serum albumin (BSA) with azido-4,5-diethynyl-1*H*-1,2,3-triazole **8a** using Cu-catalyzed azide-alkyne cycloaddition (CuAAC). 

For this purpose, the BSA protein was modified with *N*-propargylmaleimide (PM). Modified BSA (BSA-PM) was prepared by the protocol based on the interaction of the maleimide fragment with thiol groups of protein [30]. Then, the obtained conjugate BSA-PM was involved in the copper-catalyzed azide–alkyne cycloaddition using sodium ascorbate as a Cu(II) reducing agent and tris((3-hydroxypropyltriazolyl)methyl)amine (THPTA) as a ligand which accelerates the reaction and serves as a protecting source for biomolecules from oxidation [31]. Premodified BSA was purified by gel chromatography (Figure 8). A bright blue fluorescence observed under UV irradiation obviously indicated the moving of the labeled BSA through the column and the presence of labeled BSA in the collected fractions. Then, the absorption spectra for the collected fractions were recorded (Figure 8), which made it possible to detect the labeled protein in two main fractions (fr. 9, 10). 

The right graph shows the spectra of the reference compounds: UV spectra of 50 μM of BSA solution in buffer, dye **8a** in buffer solution, and a mixture of BSA and **8a** solutions. 

Comparing the absorption spectra of fractions 9 and 10 with covalently labeled albumin with the reference compounds clearly showed that in the spectrum of the labeled protein, 3 maxima appear: at 280 nm, which is characteristic of BSA, and at 330 nm and 450 nm, which are present in dye **8a**.

Figure 9A shows the electropherograms of the **8a**-labeled protein fractions that appeared when irradiated with soft ultraviolet light (365 nm). A sample containing pure protein was not detected because BSA does not have its own fluorescence. After UV visualization, the gel chromatogram was stained with Coomassie dye (Figure 9B) to confirm the presence of pure and bound proteins in the experimental samples.

The BSA was successfully labeled with NSC-triazole **8a**. Moreover, to our surprise, many non-specific, non-covalent hydrophobic bindings of the triazole dye **8a** to BSA were found. Thus, in the UV-stained electropherogram, along with the labeled BSA, the presence of free dye molecules was observed. This finding opens the way for further possible applications of the non-covalent labeling of proteins with triazole-based fluorescent dyes.

Bioimaging in Living Cells. The fluorescent imaging of living systems has become an important tool for studying biological phenomena both in vitro and in vivo. Multifunctional fluorescent dyes are vital for promoting the development of bioimaging technologies.

Biological material, as a rule, fluoresces extremely weakly by itself. However, due to bright and diverse fluorescent molecules (fluorophores) capable of specifically staining different structures of tissues and cells, it is possible to visualize many biological objects.

We performed some experiments to evaluate synthesized fluorophores during the imaging of cells. HeLa cells were cultured in media with fluorescent dyes (**5d**) for only 1 h and then analyzed by confocal laser scanning microscopy (Figure 10a). We found that 4,5-bis(arylethynyl)triazoles effectively penetrated into the cells and gave a blue glow to the cytoplasm of the cells. 

### 2.4. The Study of 4,5-diarylethynyl-1H-1,2,3-triazole Cytotoxycity

HeLa and HEK293 cell lines are widely used for cytotoxicity studies of a wide variety of compounds, ranging from plant extracts [32] to nanoparticles [33]. The MTT test allow for the assessment of the cytotoxicity of 4,5-bis(arylethynyl)triazoles **5a**–**m** and **8b, 9a** on two distinct cell lines: HEK293 and HeLa. The examined compounds did not show any significant cytotoxic effect on either cell line at concentrations lower than 50 µM (Figure 11). The greatest cytotoxic effect for both the HEK293 cell line and HeLa cells was shown by triazole **9a** containing an isothiocyanate group. Compound **5i** was cytotoxic for HEK293 cells.

In summary, the primary screening of the biological properties showed the low toxicity of the obtained 4,5-diethynyl-1*H*-1,2,3-triazoles, which makes them promising candidates for the further development of fluorescent labels for cytological studies.

## 3. Materials and Methods

### 3.1. General Information

Solvents and reagents used for reactions were purchased from commercial suppliers. Catalyst Pd(PPh_3_)_4_ was purchased from Sigma-Aldrich (München, Germany). Solvents were dried under standard conditions; chemicals were used without further purification. CuI(PPh_3_)_3_ [34], 1-iodobuta-1,3-diynes **1a**–**d,** ethyl 2-azidoacetate **2** [24], and 3-azidopropan-1-amine **7** [35] were synthesized using known procedures. Evaporation of solvents and concentration of reaction mixtures were performed in vacuum at 35 °C on a rotary evaporator. Thin-layer chromatography (TLC) was carried out on silica gel plates (Silica gel 60, F254, Merck, Darmstadt, Germany) with detection by UV. ^1^H and ^13^C NMR spectra (see Appendix A) were recorded at 400 and 100 MHz or 126 MHz, respectively, at 25 °C in CDCl_3_ without the internal standard, using a 400 MHz Avance spectrometer and 500 MHz Bruker Avance III (Bruker, Billerica, MA, USA). The ^1^H-NMR data were reported as chemical shifts (δ), multiplicity (s, singlet; d, doublet; t, triplet; q, quartet; m, multiplet), coupling constants (*J*, given in Hz), and number of protons. Chemical shifts for ^1^H and ^13^C were reported as values (ppm) and referenced to residual solvent (δ = 7.26 ppm for ^1^H; δ = 77.16 ppm for ^13^C–for spectra in CDCl_3_). High resolution mass spectra (HRMS) were determined using electrospray ionization (ESI) in the mode of positive ion registration with a Bruker micrOTOF mass analyzer (Billerica, MA, USA). UV–Vis spectra for solutions of all compounds were recorded on a UV-1800 spectrophotometer (Shimadzu, Kyoto, Japan) at room temperature. Fluorescence spectra for the same solutions were recorded on a FluoroMax-4 spectrofluorometer (Horiba Scientific, Glasgow, Scotland) at room temperature. The single-crystal X-ray diffraction studies were carried out on a diffractometer at 100 K using Cu Kα radiation (λ = 1.54180 Å). Using Olex2 [36], the structure was solved with the SUPERFLIP structure solution program [37] using charge flipping and refined with the SHELXL refinement package [38] using least-squares minimization. 

### 3.2. Synthetic Methods and Analytic Data of Compounds

#### 3.2.1. General Procure for the CuAAC

An azide (1.00 equiv.), CuI(PPh_3_)_3_ (5 mol%), and 2,6-lutidine (4 mol%) were consistently added in a screw vial to 1-iodobuta-1,3-diyne (1.00 equiv.). The thick resulting mixture was vigorously stirred for 5−24 h at room temperature. After completion of the reaction (TLC control), the reaction mixture was diluted with CH_2_Cl_2_ and a saturated aqueous solution of NH_4_Cl. The reaction mixture was shaken; the organic layer was separated, dried over anhydrous Na_2_SO_4_, and concentrated under reduced pressure to yield the crude product, which was purified by column chromatography on silica gel.

Ethyl 2-{4-[(4-cyanophenyl)ethynyl]-5-iodo-1*H*-1,2,3-triazol-1-yl}acetate (**3a**) was prepared in accordance with the general procedure from ethyl 2-azidoacetate **2** (105.1 mg, 0.81 mmol) and iodoalkyne **1a** (225.6 mg, 0.81 mmol). Reaction time: 19 h. The crude product was purified by column chromatography (eluent: hexane/EtOAc = 3:1) to afford a beige solid (233 mg, 72%). ^1^H NMR (CDCl_3_, 400 MHz) δ 7.66–7.59 (m, 4H, Ar), 5.21 (s, 2H, CH_2_), 4.30 (q, *J* = 7.1 Hz, 2H, CH_2_), 1.32 (t, *J* = 7.1 Hz, 3H, CH_3_). ^13^C NMR (CDCl_3_, 101 MHz) δ 165.1, 138.1, 132.4, 132.3, 127.0, 118.4, 112.6, 93.3, 86.3, 82.5, 62.9, 51.9, 14.2. HRMS ESI: [M + Na]^+^ calcd for C_15_H_11_N_4_O_2_Na^+^: 428.9819; found: 428.9821. 

Ethyl 2-{4-[(4-chlorophenyl)ethynyl]-5-iodo-1*H*-1,2,3-triazol-1-yl}acetate (**3b**) was prepared in accordance with the general procedure from ethyl 2-azidoacetate **2** (202.8 mg, 1.57 mmol) and iodoalkyne **1b** (450 mg, 1.57 mmol). Reaction time: 15 h. The crude product was purified by column chromatography (eluent: hexane/EtOAc = 5:1) to afford a beige solid (522 mg, 80%). ^1^H NMR (CDCl_3_, 400 MHz) δ 7.55–7.48 (m, 2H, Ar), 7.38–7.32 (m, 2H, Ar), 5.20 (s, 2H, CH_2_), 4.30 (q, *J* = 7.1 Hz, 2H, CH_2_), 1.31 (t, *J* = 7.1 Hz, 3H, CH_3_). ^13^C NMR (CDCl_3_, 101 MHz) δ 165.2, 138.6, 135.4, 133.2, 129.0, 120.7, 94.1, 85.7, 79.3, 62.9, 51.9, 14.2. HRMS ESI: [M + Na]^+^ calcd. for C_14_H_11_ClN_3_IO_2_Na^+^: 437.9477; found 437.9493.

Ethyl 2-{4-[(4-(dimethylamino)phenyl)ethynyl]-5-iodo-1*H*-1,2,3-triazol-1-yl}acetate (**3c**) was prepared in accordance with the general procedure from ethyl 2-azidoacetate **2** (123.8 mg, 0.96 mmol) and iodoalkyne **1c** (283 mg, 0.96 mmol). Reaction time: 21 h. The crude product was purified by column chromatography (eluent: hexane/EtOAc = 3:1) to afford a beige solid (283 mg, 70%). ^1^H NMR (CDCl_3_, 400 MHz) δ 7.46 (d, *J* = 8.9 Hz, 2H, Ar), 6.66 (d, *J* = 8.9 Hz, 2H, Ar), 5.18 (s, 2H, CH_2_), 4.28 (q, *J* = 7.1 Hz, 2H, CH_2_), 3.01 (s, 6H, 2CH_3_), 1.30 (τ, *J* = 7.1 Hz, 3H, CH_3_). ^13^C NMR (CDCl_3_, 101 MHz) δ 165.3, 150.7, 139.6, 133.2, 111.8, 108.7, 96.7, 84.7, 76.2, 62.7, 51.9, 40.3, 14.2. HRMS ESI: [M + Na]^+^ calcd for C_16_H_17_N_4_O_2_Na^+^: 447.0302; found: 447.0288.

Ethyl 2-{5-iodo-4-[ (4-nitrophenyl)ethynyl]-1*H*-1,2,3-triazol-1-yl}acetate (**3d**) was prepared in accordance with the general procedure from ethyl 2-azidoacetate **2** (28 mg, 0.22 mmol) and iodoalkyne **1d** (64 mg, 0.22 mmol). The reaction mixture was stirred at 40 °C. Reaction time: 4 h. The crude product was purified by column chromatography (eluent: hexane/EtOAc = 3:1) to afford a light-brown solid (60 mg, 65%). ^1^H NMR (CDCl_3_, 400 MHz) δ 8.28–8.21 (m, 2H, Ar), 7.78–7.70 (m, 2H, Ar), 5.22 (s, 2H, CH_2_), 4.30 (q, *J* = 7.2 Hz, 2H, CH_2_), 1.32 (t, *J* = 7.1 Hz, 3H, CH_3_). ^13^C NMR (CDCl_3_, 101 MHz) δ 164.9, 147.6, 137.8, 132.5, 128.8, 123.7, 92.9, 86.4, 83.2, 62.8, 51.8, 14.1. HRMS ESI: [M + Na]^+^ calcd for C_16_H_17_N_4_O_2_Na^+^: 448.9717; found: 448.9703.

#### 3.2.2. General Procure for the Sonogashira Cross-Coupling

5-Iodo-1*H*-1,2,3-triazoles **3a**−**d** (1 equiv.), CuI (10 mol%), K_3_PO_4_ (1.1 equiv.), and Pd(PPh_3_)_4_ (5 mol%) were placed in a vial. The vial was sealed, and the mixture was evacuated and flushed with Ar several times. THF (1 mL) was added, and the mixture was stirred at room temperature for 10 min; then, an alkyne **4a**−**f** (1.1–2 equiv.) was added. The vial with reaction mixture was placed in a pre-heated oil bath (65 °C) and stirred at this temperature for 1–11 h (TLC control). After cooling to room temperature, the reaction mixture was filtered through a silica gel pad, and the pad was washed with CH_2_Cl_2_ (3 × 10 mL). The solvents were removed under reduced pressure, and the crude product was purified by column chromatography on silica gel.

Ethyl 2-{4-[(4-cyanophenyl)ethynyl]-5-[(2-(dimethylamino)phenyl)ethynyl]-1*H*-1,2,3-triazol-1-yl}acetate (**5a**) was prepared in accordance with the general procedure from 5-iodo-1*H*-1,2,3-triazole **5a** (200 mg, 0.49 mmol), 2-ethynyl-*N,N*-dimethylaniline (71.5 mg, 0.49 mmol), K_3_PO_4_ (115 mg, 0.54 mmol), CuI (9.4 mg, 0.049 mmol), and Pd(PPh_3_)_4_ (28.5 mg, 0.02 mmol); reaction time: 11 h. The crude product was purified by chromatography (eluent: hexane/EtOAc = 5:1) to afford a yellow-green oil (173 mg, 83%). ^1^H NMR (CDCl_3_, 400 MHz) δ 7.66–7.64 (m, 4H, Ar), 7.46–7.43 (m, 1H, Ar), 7.36–7.32 (m, 1H, Ar), 6.95–6.89 (m, 2H, Ar), 5.24 (s, 2H, CH_2_), 4.28 (q, *J* = 7.1 Hz, 2H, CH_2_), 2.97 (s, 6H, 2CH_3_), 1.28 (t, *J* = 7.1 Hz, 3H, CH_3_). ^13^C NMR (CDCl_3_, 101 MHz) δ 165.4, 155.5, 134.8, 132.4, 132.28, 132.26, 131.5, 127.3, 126.2, 120.5, 118.4, 117.3, 112.4, 111.9, 104.5, 93.5, 82.8, 62.7, 50.1, 43.7, 14.2. HRMS ESI: [M + Na]^+^ calcd. for C_25_H_21_N_5_NaO_2_^+^: 446.1587; found: 446.1593. 

Ethyl 2-{4-[(4-chlorophenyl)ethynyl]-5-[(2-(dimethylamino)phenyl)ethynyl]-1*H*-1,2,3-triazol-1-yl}acetate (**5b**) was prepared in accordance with the general procedure from 5-iodo-1*H*-1,2,3-triazole **3b** (138 mg, 0.33 mmol), 2-ethynyl-*N,N*-dimethylaniline (48.2 mg, 0.33 mmol), K_3_PO_4_ (77.5 mg, 0.37 mmol), CuI (6.3 mg, 0.033 mmol), and Pd(PPh_3_)_4_ (19.2 mg, 0.017 mmol); reaction time: 5 h. The crude product was purified by chromatography (eluent: hexane/acetone = 5:1) to afford a yellow oil (83 mg, 58%). ^1^H NMR (CDCl_3_, 400 MHz) δ 7.53–7.47 (m, 2H, Ar), 7.44 (dd, *J* = 7.7, 1.7 Hz, 1H, Ar), 7.36–7.30 (m, 3H, Ar), 6.87 (m, 2H, Ar), 5.23 (s, 2H, CH_2_), 4.28 (q, *J* = 7.1 Γц, 2H, CH_2_), 2.97 (s, 6H, 2CH_3_), 1.28 (t, *J* = 7.1 Hz, 3H, CH_3_). ^13^C NMR (CDCl_3_, 101 MHz) δ 165.6, 155.5, 135.2, 134.7, 133.1, 133.05, 131.3, 129.0, 125.6, 120.9, 120.4, 117.3, 112.1, 104.1, 94.3, 79.5, 77.7, 77.4, 62.7, 50.1, 43.7, 31.1, 14.2. HRMS ESI: [M + H]^+^ calcd. for C_24_H_22_ClN_4_O_2_^+^: 433.1426; found: 433.1425.

Ethyl 2-{4-[(4-cyanophenyl)ethynyl]-5-[(4-(dimethylamino)phenyl)ethynyl]-1*H*-1,2,3-triazol-1-yl}acetate (**5c**) was prepared in accordance with the general procedure from 5-iodo-1*H*-1,2,3-triazole **3a** (115 mg, 0.28 mmol), 4-ethynyl-*N,N*-dimethylaniline (41.1 mg, 0.28 mmol), K_3_PO_4_ (66.1 mg, 0.31 mmol), CuI (5.4 mg, 0.028 mmol), and Pd(PPh_3_)_4_ (16.4 mg, 0.014 mmol); reaction time: 3 h. The crude product was purified by chromatography (eluent: benzene/acetone = 100:1) to afford a beige solid (106 mg, 89%). ^1^H NMR (CDCl_3_, 400 MHz, δ)7.69–7.61 (m, 4H, Ar), 7.40 (d, *J* = 8.9 Hz, 2H, Ar), 6.66 (d, *J* = 8.9 Hz, 2H, Ar), 5.21 (s, 2H, CH_2_), 4.28 (q, *J* = 7.1 Hz, 2H, CH_2_), 3.03 (s, 6H, 2CH_3_), 1.28 (t, *J* = 7.1 Γц, 3H, CH_3_). ^13^C NMR (CDCl_3_, 101 MHz) δ 165.6, 151.4, 133.34, 132.32, 132.2, 131.9, 127.5, 126.5, 118.6, 112.2, 111.8, 106.7, 106.6, 93.4, 83.0, 71.0, 62.7, 50.2, 40.2, 14.2. HRMS ESI: calcd. for [M + Na]^+^ C_25_H_21_N_5_NaO_2_^+^: 446.1587; found: 446.1587. IR, cm^−1^, *ν*: 2223 and 2202 (C≡C), 1748 (CO).

Ethyl 2-{4-[(4-chlorophenyl)ethynyl]-5-[(4-(dimethylamino)phenyl)ethynyl]-1*H*-1,2,3-triazol-1-yl}acetate (**5d**) was prepared in accordance with the general procedure from 5-iodo-1H-1,2,3-triazole **3b** (138 mg, 0.33 mmol), 4-ethynyl-*N,N*-dimethylaniline (48.2 mg, 0.33 mmol), K_3_PO_4_ (77.5 mg, 0.37 mmol), CuI (6.3 mg, 0.033 mmol), and Pd(PPh_3_)_4_ (19.2 mg, 0.017 mmol); reaction time: 5 h. The crude product was purified by chromatography (eluent: hexane/acetone = 5:1) to afford light-yellow crystals (89 mg, 62% (benzene)). ^1^H NMR (CDCl_3_, 400 MHz) δ 7.56–7.49 (m, 2H, Ar), 7.46–7.31 (m, 4H, Ar), 6.67–6.65 (m, 2H, Ar), 5.20 (s, 2H, CH2), 4.28 (q, *J* = 7.2 Hz, 2H, CH_2_), 3.03 (s, 6H, 2CH_3_), 1.28 (t, *J* = 7.2 Hz, 3H, CH_3_). ^13^C NMR (CDCl_3_, 101 MHz) δ 165.7, 151.3, 135.1, 133.3, 133.1, 132.5, 128.9, 125.9, 121.1, 111.8, 107.0, 106.1, 94.1, 79.6, 71.2, 62.6, 50.1, 40.2, 14.2. HRMS ESI: [M + Na]^+^ calcd. for C_24_H_21_ClN_4_O_2_Na^+^: 455.1245; found: 455.1248. Appropriate crystals for X-ray analyses were obtained from chloroform solution. Crystallographic data for **5d** were deposited with the Cambridge Crystallographic Data Centre, no. CCDC 2168522.

Ethyl 2-{4-[(4-cyanophenyl)ethynyl]-5-[(4-methoxyphenyl)ethynyl]-1*H*-1,2,3-triazol-1-yl}acetate (**5e**) was prepared in accordance with the general procedure from 5-iodo-1*H*-1,2,3-triazole **3a** (115 mg, 0.028 mmol), 1-ethynyl-4-methoxybenzene (37.4 mg, 0.028 mmol), K_3_PO_4_ (66.1 mg, 0.031 mmol), CuI (5.4 mg, 0.028 mmol), and Pd(PPh_3_)_4_ (16.4 mg, 0.014 mmol); reaction time: 3 h. The crude product was purified by chromatography (eluent: benzene) to afford a yellow oil (84 mg, 72%). ^1^H NMR (CDCl_3_, 400 MHz) δ 7.70–7.62 (m, 4H, Ar), 7.51–7.47 (m, 2H, Ar), 6.95–6.90 (m, 2H, Ar), 5.22 (s, 2H, CH_2_), 4.29 (q, *J* = 7.1 Hz, 2H, CH_2_), 3.86 (s, 3H, OCH_3_), 1.28 (t, *J* = 7.1 Hz, 3H, CH_3_). ^13^C NMR (CDCl_3_, 101 MHz) δ 165.5, 161.3, 133.7, 132.5, 132.4, 132.3, 127.3, 125.9, 118.5, 114.6, 112.7, 112.4, 104.7, 93.6, 82.6, 71.5, 62.7, 55.6, 50.2, 14.2. HRMS ESI: [M + Na]^+^ calcd. for C_24_H_18_N_4_NaO_3_^+^: 433.1271; found: 433.1270. 

Ethyl 2-{4-((4-chlorophenyl)ethynyl)-5-((2-methoxyphenyl)ethynyl)-1*H*-1,2,3-triazol-1-yl)acetate (**5f**) was prepared in accordance with the general procedure from 5-iodo-1*H*-1,2,3-triazole **3b** (125 mg, 0.3 mmol), 1-ethynyl-2-methoxybenzene (56.8 mg, 0.43 mmol), K_3_PO_4_ (70.2 mg, 0.33 mmol), CuI (5.73 mg, 0.03 mmol), and Pd(PPh_3_)_4_ (27.8 mg, 0.025 mmol); reaction time: 4 h. The crude product was purified by chromatography (eluent: hexane/acetone = 5:1) to afford a beige, amorphous precipitate (73 mg, 58% (benzene)). ^1^H NMR (CDCl_3_, 400 MHz) δ 7.57–7.52 (m, 2H, Ar), 7.49 (dd, *J* = 7.6, 1.7 Hz, 1H, Ar), 7.45–7.34 (m, 3H, Ar), 7.02–6.91 (m, 2H, Ar), 5.27 (s, 2H, CH_2_), 4.27 (t, *J* = 7.1 Hz, 2H, CH_2_), 3.90 (s, 3H, OCH_3_), 1.26 (t, *J* = 7.1 Hz, 3H, CH_3_). ^13^C NMR (CDCl_3_, 101 MHz) δ 165.6, 160.7, 135.2, 133.3, 133.2, 132.7, 131.8, 128.9, 125.5, 121.0, 120.8, 111.0, 110.4, 100.9, 94.3, 79.4, 77.0, 62.6, 55.9, 50.1, 14.2. HRMS ESI: [M + Na]^+^ calcd. for C_23_H_18_ClN_3_O_3_Na^+^: 442.0929; found: 442.0939. 

Ethyl 2-{4-[(4-chlorophenyl)ethynyl]-5-[(4-methoxyphenyl)ethynyl]-1*H*-1,2,3-triazol-1-yl}acetate (**5g**) was prepared in accordance with the general procedure from 5-iodo-1*H*-1,2,3-triazole **3b** (138 mg, 0.33 mmol), 1-ethynyl-2-methoxybenzene (44 mg, 0.33 mmol), K_3_PO_4_ (77.5 mg, 0.37 mmol), CuI (16.6 mg, 0.03 mmol), and Pd(PPh_3_)_4_ (16.6 mg, 0.02 mmol); reaction time: 5 h. The crude product was purified by chromatography (eluent: hexane/acetone = 5:1) to afford a yellow solid (86 mg, 62%). ^1^H NMR (CDCl_3_, 400 MHz) δ 7.56–7.45 (m, 4H, Ar), 7.38–7.30 (m, 2H, Ar), 6.96–6.88 (m, 2H, Ar), 5.21 (s, 2H, CH_2_), 4.28 (q, *J* = 7.1 Hz, 2H, CH_2_), 3.88–3.83 (br.s, 3H, OCH_3_), 1.28 (t, *J* = 7.1 Hz, 3H, CH_3_). ^13^C NMR (CDCl_3_, 101 MHz) δ 165.6, 161.2, 135.2, 133.7, 133.2, 128.9, 125.3, 120.9, 114.5, 112.9, 104.3, 94.3, 79.3, 71.8, 62.7, 55.6, 50.2, 14.2. HRMS ESI: [M + Na]^+^ calcd. for C_23_H_18_ClN_3_O_3_Na^+^: 442.0934; found: 442.0939. 

Ethyl 2-{5-[(4-cyanophenyl)ethynyl]-4-[(4-(dimethylamino)phenyl)ethynyl]-1*H*-1,2,3-triazol-1-yl}acetate (**5h**) was prepared in accordance with the general procedure from 5-iodo-1*H*-1,2,3-triazole **5c** (140 mg, 0.033 mmol), 4-ethynylbenzonitrile (52.9 mg, 0.33 mmol), K_3_PO_4_ (77.05 mg, 0.36 mmol), CuI (6.3 mg, 0.033 mmol), and Pd(PPh_3_)_4_ (19.1 mg, 0.016 mmol); reaction time: 5.5 h. The crude product was purified by chromatography (eluent: benzene/acetone = 100:1) to afford a light-brown solid (100 mg, 72%). ^1^H NMR (CDCl_3_, 400 MHz) δ 7.68 (d, *J* = 8.3 Hz, 2H, Ar), 7.63 (d, *J* = 8.3 Hz, 2H, Ar), 7.45 (d, *J* = 8.8 Hz, 2H, Ar), 6.66 (d, *J* = 8.8 Hz, 2H, Ar), 5.21 (s, 2H, CH_2_), 4.28 (q, *J* = 7.1 Hz, 2H, CH_2_), 3.01 (s, 6H, 2CH_3_), 1.28 (t, *J* = 7.1 Hz, 3H, CH_3_). ^13^C NMR (CDCl_3_, 101 MHz) δ 165.5, 150.8, 135.7, 133.2,132.4, 132.3, 126.0, 122.9, 118.2, 113.3, 111.8, 108.4, 101.2, 98.1, 75.9, 62.8, 50.4, 40.2, 14.2. HRMS ESI: [M + Na]^+^ calcd. for C_25_H_21_N_5_NaO_2_^+^: 446.1587; found: 446.1577. 

Methyl 4-{(4-[(4-(dimethylamino)phenyl)ethynyl]-1-(2-ethoxy-2-oxoethyl)-1*H*-1,2,3-triazol-5-yl)ethynyl}benzoate (**5i**) was prepared in accordance with the general procedure from 5-iodo-1*H*-1,2,3-triazole **3c** (140 mg, 0.33 mmol), methyl 4-ethynylbenzoate (42 mg, 0.33 mmol), K_3_PO_4_ (77.05 mg, 0.36 mmol), CuI (6.3 mg, 0.033 mmol), and Pd(PPh_3_)_4_ (19.1 mg, 0.016 mmol); reaction time: 5.5 h. The crude product was purified by chromatography (eluent: benzene/acetone = 50:1) to afford a yellow solid (100 mg, 72% (benzene)). ^1^H NMR (CDCl_3_, 400 MHz) δ 8.06 (d, *J* = 8.3 Hz, 2H, Ar), 7.61 (d, *J* = 8.3 Hz, 2H, Ar), 7.47 (d, *J* = 8.9 Hz, 2H, Ar), 6.66 (d, *J* = 8.9 Hz, 2H, Ar), 5.22 (s, 2H, CH_2_), 4.28 (q, *J* = 7.1 Hz, 2H, CH_2_), 3.94 (s, 3H, COOCH_3_), 3.00 (s, 6H, 2CH_3_), 1.28 (t, *J* = 7.1 Hz, 3H, CH_3_). ^13^C NMR (CDCl_3_, 101 MHz) δ 166.3, 165.6, 150.8, 135.4, 133.2, 131.8, 131.1, 129.9, 125.7, 123.4, 111.8, 108.6, 102.4, 97.8, 76.0, 76.0, 62.7, 52.5, 50.3, 40.3, 14.2. HRMS ESI: [M + Na]^+^ calcd. for C_26_H_24_N_4_NaO_4_^+^: 479.1690; found: 479.1698. IR, cm^−1^, *ν*: 2208 (C≡C), 1751 (CO), 1721 (CO).

Ethyl 2-{5-[(2-(dimethylamino)phenyl)ethynyl]-4-[(4-(dimethylamino)phenyl)ethynyl]-1*H*-1,2,3-triazol-1-yl}acetate (**5j**) was prepared in accordance with the general procedure from 5-iodo-1*H*-1,2,3-triazole **3c** (55 mg, 0.13 mmol), 2-ethynyl-*N,N*-dimethylaniline (18 mg, 0.13 mmol), K_3_PO_4_ (30.3 mg, 0.14 mmol), CuI (2.5 mg, 0.013 mmol), and Pd(PPh_3_)_4_ (7.5 mg, 0.007 mmol); reaction time: 3 h. The crude product was purified by chromatography (eluent: hexane/acetone = 3:1) to afford a white solid (32 mg, 57%). ^1^H NMR (CDCl_3_, 400 MHz) δ 7.48–7.39 (m, 3H, Ar), 7.31–7.28 (m, 1H, Ar), 6.95–6.84 (m, 2H, Ar), 6.70–6.61 (m, 2H, Ar), 5.21 (s, 2H, CH_2_), 4.27 (q, *J* = 7.1 Hz, 2H, CH_2_), 3.0 (s, 6H, 2CH_3_), 2.99 (s, 6H, 2CH_3_), 1.27 (t, *J* = 7.1 Hz, 3H, CH_3_). ^13^C NMR (CDCl_3_, 126 MHz) δ 165.8, 155.4, 150.6, 134.7, 134.4, 133.1, 131.0, 124.6, 120.3, 117.2, 112.4, 111.9, 109.0, 103.6, 97.0, 78.3, 76.5, 62.6, 50.1, 43.7, 40.3, 14.2. HRMS ESI: [M + H]^+^ calcd. for C_26_H_28_N_5_O_2_^+^: 442.2238; found: 442.2234.

Ethyl 2-{5-[(2-(dimethylamino)phenyl)ethynyl]-4-[(4-nitrophenyl)ethynyl]-1*H*-1,2,3-triazol-1-yl}acetate (**5k**) was prepared in accordance with the general procedure from 5-iodo-1*H*-1,2,3-triazole **3d** (58 mg, 0.14 mmol), 2-ethynyl-*N,N*-dimethylaniline (20 mg, 0.14 mmol), K_3_PO_4_ (32 mg, 0.15 mmol), CuI (2.6 mg, 0.014 mmol), and Pd(PPh_3_)_4_ (7.9 mg, 0.007 mmol); reaction time: 3 h. The crude product was purified by chromatography (eluent: hexane/acetone = 3:1) to afford a dark-orange solid (42 mg, 69%). ^1^H NMR (CDCl_3_, 400 MHz) δ 8.26–8.20 (m, 2H, Ar), 7.74–7.68 (m, 2H, Ar), 7.45 (dd, *J* = 7.6, 1.7 Hz, 1H, Ar), 7.37–7.31 (m, 1H, Ar), 6.98–6.87 (m, 2H, Ar), 5.25 (s, 2H, CH_2_), 4.28 (q, *J* = 7.1 Hz, 2H, CH_2_), 2.98 (s, 6H, N(CH_3_)_2_), 1.29 (t, *J* = 7.1 Hz, 3H, CH_3_). ^13^C NMR (101 MHz, CDCl_3_) δ 165.5, 155.5, 147.6, 134.8, 132.5, 132.3, 131.5, 129.2, 126.3, 123.9, 120.5, 117.3, 111.9, 104.7, 93.3, 83.6, 77.37, 62.7, 50.1, 43.7, 14.2. HRMS ESI: [M + H]^+^ calcd. for C_24_H_22_N_5_O_4_^+^: 444.1666; found: 444.1662. 

Methyl 2-{(4-[(4-cyanophenyl)ethynyl]-1-(2-ethoxy-2-oxoethyl)-1*H*-1,2,3-triazol-5-yl)ethynyl}benzoate (**5l**) was prepared in accordance with the general procedure from 5-iodo-1*H*-1,2,3-triazole **3a** (100.0 mg, 0.246 mmol), methyl 2-ethynylbenzoate (39.5 mg, 0.246 mmol), K_3_PO_4_ (57.3 mg, 0.27 mmol), CuI (5.0 mg, 0.025 mmol), and Pd(PPh_3_)_4_ (14.2 mg, 0.012 mmol); reaction time: 40 min. The crude product was purified by chromatography (eluent: benzene) to afford a yellow solid (71 mg, 66%). ^1^H NMR (CDCl_3_, 400 MHz) δ 8.09–8.02 (m, 1H, Ar), 7.74–7.64 (m, 5H, Ar), 7.61–7.55 (m, 1H, Ar), 7.53–7.48 (m, 1H, Ar), 5.44 (s, 2H, CH_2_), 4.27 (m, 2H, CH_2_), 3.91 (s, 3H, COOCH_3_), 1.27 (m, 3H, CH_3_). ^13^C NMR (CDCl_3_, 101 MHz) δ 165.8, 165.6, 134.3, 132.7, 132.4, 132.3, 132.2, 131.8, 130.9, 129.8, 127.3, 125.8, 121.8, 118.5, 112.4, 102.8, 93.7, 82.6, 62.5, 52.4, 50.3, 14.2. HRMS ESI: [M + H]^+^ calcd. for C_25_H_19_N_4_O_4_^+^: 439.1401; found: 439.1401. 

Ethyl 2-{4-[(4-cyanophenyl)ethynyl]-5-[(1,3,5-trimethyl-1*H*-pyrazol-4-yl)ethynyl]-1H-1,2,3-triazol-1-yl}acetate (**5m**) was prepared in accordance with the general procedure from 5-iodo-1*H*-1,2,3-triazole **3b** (100.0 mg, 0.246 mmol), methyl 2-ethynylbenzoate (39.5 mg, 0.246 mmol), K_3_PO_4_ (57.3 mg, 0.27 mmol), CuI (5.0 mg, 0.025 mmol), and Pd(PPh_3_)_4_ (14.2 mg, 0.012 mmol); reaction time: 40 min. The crude product was purified by chromatography (eluent: benzene) to afford a white solid (71 mg, 66%). ^1^H NMR (CDCl_3_, 400 MHz) δ 7.65–7.60 (m, 4H, Ar), 5.19 (s, 2H, CH_2_), 4.27 (q, *J* = 7.1 Hz, 2H, CH_2_), 3.74 (s, 3H, NCH_3_), 2.33 (s, 3H, CH_3_), 2.28 (s, 3H, CH_3_), 1.27 (t, *J* = 7.1 Hz, 3H, CH_3_). ^13^C NMR (CDCl_3_, 101 MHz) δ 165. 5, 150.2, 143.4, 132.2, 132.2, 132.1, 127.3, 126.4, 118.4, 112.3, 99.9, 97.7, 93.4, 82.7, 76.4, 62.7, 50.1, 36.4, 14.2, 12.5, 10.7. HRMS ESI: [M + H]^+^ calcd. for C_23_H_21_N_6_O_2_^+^: 413.1721; found: 413.1721. 

#### 3.2.3. General Procure for the Synthesis of Triazoloacids **7**

To a stirred solution of triazoles **5** (1 equiv.) in 15 mL of THF an aqueous solution of LiOH (2–4 equiv., 0.05 M) was added, and the reaction mixture was stirred at room temperature for 15 h; then, a 1% aqueous solution of HCl was added up to a pH ~4–5, followed by extraction with EtOAc (3 × 15 mL). The combined extracts were washed with brine (1 × 10 mL) and dried with Na_2_SO_4_. After removal of the solvent under reduced pressure, the product obtained was a light-yellow powder, which we used without additional purification in the next step.

2-{4-[(4-Chlorophenyl)ethynyl)-5-((2-(dimethylamino)phenyl)ethynyl]-1*H*-1,2,3-triazol-1-yl}acetic acid **7a** was prepared in accordance with the general procedure from **5b** (150 mg, 0.35 mmol), LiOH×H_2_O (29 mg, 0.69 mmol). The reaction gave the acid **7a** as a yellow solid (103 mg, 80%). 

2-{5-[(4-Cyanophenyl)ethynyl]-4-[(4-(dimethylamino)phenyl)ethynyl]-1*H*-1,2,3-triazol-1-yl}acetic acid **7b** was prepared in accordance with the general procedure from **5b** (65 mg, 0.15 mmol), LiOH×H_2_O (25 8 mg, 0.61 mmol). The reaction gave the acid **7b** as a dark-yellow solid (48.7 mg, 80%). 

#### 3.2.4. General Procure for the Synthesis of Amidoazides **8**

To 0.05 M well-degassed, stirred solution of **7** (1 equiv.) in THF *N,N*-diisopropylethylamine (DIPEA, 2 equiv.), 3-azidopropan-1-amine (1 equiv.), and 1-[bis(dimethylamino)methylene]-1H-1,2,3-triazolo [4,5-b]pyridinium 3-oxide hexafluorophosphate (HATU, 1 equiv.) were added at 0 °C. The reaction mixture was stirred at room temperature for 15 h, then diluted with EtOAc (10 mL), and washed with a saturated aqueous solution of NH_4_Cl (3 × 5 mL). The organic layer was then washed with a water (2 × 5 mL) and brine (1 × 5 mL) solution, dried over anhydrous Na_2_SO_4_, and concentrated under reduced pressure to yield the crude product, which was purified by column chromatography.

N-(3-Azidopropyl)-2-{4-[(4-chlorophenyl)ethynyl)-5-((2-(dimethylamino)phenyl)ethynyl]-1*H*-1,2,3-triazol-1-yl}acetamide **8a** was prepared in accordance with the general procedure from **7a** (56 mg, 0.14 mmol), 3-azidopropan-1-amine (13.8 mg, 0.14 mmol), DIPEA (35.8 mg, 0.277 mmol), and HATU (57.8 mg, 0.152 mmol); reaction time: 15 h. The crude product was purified by chromatography (eluent: hexane/acetone = 3:1) to afford a yellow solid (49 mg, 73%). ^1^H NMR (CDCl_3_, 400 MHz) δ 7.44–7.42 (m, 3H, Ar), 7.34–7.25 (m, 3H, Ar), 6.94–6.88 (m, 2H, Ar), 6.66–6.64 (m, 1H, NH), 5.16 (s, 2H, CH_2_), 3.41 (q, *J* = 6.4 Hz, 2H, CH_2_), 3.33 (t, *J* = 6.6 Hz, 2H, CH_2_), 2.95 (s, 6H, N(CH_3_)_2_), 1.79 (p, *J* = 6.6 Hz, 2H, CH_2_). ^13^C NMR (CDCl_3_, 101 MHz) δ 164.8, 155.5, 135.4, 134.9, 133.3, 133.0, 131.5, 129.0, 125.8, 120.6, 117.3, 111.7, 105.2, 94.8, 78.9, 77.2, 52.1, 49.4, 43.7, 37.7, 28.6. HRMS ESI: [M + H]^+^ calcd. for C_25_H_24_ClN_8_O^+^: 487,1756; found: 487.1763.

N-(3-Azidopropyl)-2-{5-[(4-cyanophenyl)ethynyl]-4-[(4-(dimethylamino)phenyl)ethynyl]-1*H*-1,2,3-triazol-1-yl}acetamide **8b** was prepared in accordance with the general procedure from **7b** (48 mg, 0.12 mmol), 3-azidopropan-1-amine (12.2 mg, 0.12 mmol), DIPEA (47.1 mg, 0.36 mmol), and HATU (46.2 mg, 0.12 mmol); reaction time: 3 h. The crude product was purified by chromatography (eluent: DCM/methanol = 50:1) to afford a yellow solid (39 mg, 67%). ^1^H NMR (CDCl_3_, 400 MHz) δ 7.65–7.60 (m, 2H, Ar), 7.58–7.52 (m, 2H, Ar), 7.44–7.38 (m, 2H, Ar), 6.71 (t, *J* = 5.9 Hz, 1H, NH), 6.61–6.53 (m, 2H, Ar), 5.15 (s, 2H, CH_2_), 3.44 (q, *J* = 6.5 Hz, 2H, CH_2_), 3.36 (t, *J* = 6.6 Hz, 2H, CH_2_), 3.00 (s, 6H, N(CH_3_)_2_), 1.82 (p, *J* = 6.6 Hz, 2H, CH_2_). ^13^C NMR (CDCl_3_,101 MHz) δ 164.8, 150.8, 135.9, 133.2, 132.4, 132.3, 125.6, 123.2, 118.2, 113.3, 111.7, 108.1, 102.2, 98.7, 76.6, 75.4, 52.5, 49.4, 40.2, 37.8, 28.7. HRMS ESI: [M + H]^+^ calcd. for C_26_H_24_N_9_O^+^: 478.2098; found: 478.2099.

2-{4-[(4-Chlorophenyl)ethynyl]-5-[(2-(dimethylamino)phenyl)ethynyl]-1*H*-1,2,3-triazol-1-yl}-N-(3-isothiocyanatopropyl)acetamide **9a.** To a stirred solution of azide **8a** (50 mg, 0.103 mmol) in THF (0.5 mL) were added carbon disulfide (10.9 mg, 0.144 mmol) and PPh_3_ (32.3 mg, 0.123 mmol). The reaction mixture was stirred at room temperature for 7 h (TLC-control). The crude product was purified by chromatography (eluent: benzene/acetone = 10:1) to afford a yellow solid (36.5 mg, 71%). ^1^H NMR (CDCl_3_, 400 MHz) δ δ 7.46–7.40 (m, 3H, Ar), 7.35 (m, 1H, Ar), 7.28 (m, 2H, Ar), 6.98–6.88 (m, 2H, Ar), 6.71–6.64 (m, 1H, NH), 5.17 (s, 2H, CH_2_), 3.54 (t, *J* = 6.6 Hz, 2H, CH_2_), 3.44 (q, *J* = 6.6 Hz, 2H, CH_2_), 1.91 (p, *J* = 6.6 Hz, 2H, CH_2_). ^13^C NMR (CDCl_3_, 101 MHz) δ 165.0, 155.4, 135.5, 134.9, 133.4, 133.0, 131.7, 129.0, 125.7, 120.8, 120.5, 117.4, 111.7, 105.2, 94.9, 89.4, 78.9, 52.2, 43.9, 42.8, 37.2, 29.9. HRMS ESI: [M + Na]^+^ calcd. for C_26_H_23_ClN_6_OSNa^+^: 525.1235; found: 525.1228.

### 3.3. The Absolute Fluorescence Quantum Yield Measurements

The absolute fluorescence quantum yield was measured on a Horiba Fluorolog-3 spectrometer (Edison, NJ, USA) equipped with an integrating sphere. A xenon lamp coupled to a double monochromator was used as the excitation light source. The sample (1 cm quartz cuvette cell with solution in THF) or blank (pure THF) was directly illuminated in the center of the integrating sphere. The optical density of all investigated sample solutions in corresponding solvents did not exceed 0.1 at the luminescence excitation wavelength. Under the same conditions (e.g., excitation wavelength, spectral resolution, and temperature), the luminescence spectrum of the sample *Ec*, the luminescence spectrum of the blank *Ea*, the Rayleigh scattering spectrum of the sample *Lc*, and the Rayleigh scattering spectrum of the solvent *La* were measured. The absolute fluorescence quantum yield was determined according to the formula:*Φ*_F_ = (*Ec* − *Ea*)/(*La* − *Lc*)(1)

### 3.4. Protein Labeling

For the separate labeling of proteins, 1 mg/mL solution of bovine serum albumin (BSA, Sigma-Aldrich, München, Germany) and 0.5 mg/mL of a solution of aldolase from rabbit muscle (Sigma-Aldrich, München, Germany) were used. The standard mixture of proteins contained BSA (Sigma-Aldrich, München, Germany), phosphorylase b, ovalbumin, carbonic anhydrase (Sigma-Aldrich, München, Germany), and recombinant KNOX-HD protein prepared according to [39], with a concentration of 0.2 mg/mL each. An amount of 40 μL of the protein solutions was mixed with 80 μL of borate buffer (100 μM, NaCl 150 μM, pH 9). An amount of 5 μL of **9b** solution (2,57 μM in DMSO) or 5 μL of DMSO (as the control without labeling) was added to the probes. Probes were incubated for 2 h at 37 °C. Excessive dye was removed using Zeba spin desalting columns (ThermoFisherScientific, Waltham, MA, USA). SDS polyacrylamide gel electrophoresis was performed according to [40]. Fluorescence was analyzed using UV transilluminator (365 nm). After that, gels were stained with Coomassie Brilliant Blue G-250 according to [41].

### 3.5. Confocal Laser Scanning Microscopy (CLSM)

HeLa cell cultures were grown in DMEM standard medium supplemented with 10% fetal bovine serum (FBS) at 37 °C in an atmosphere containing 5% CO_2_. The cells were transferred to 8-well (10,000 cells per well in 500 μL DMEM + 10% FBS) culture slides (Corning, Corning, NY, USA). The slides were incubated for 24 h, and culture medium was replaced with 500 μL DMEM + 10% FBS containing 10 μM of 5d or 5i. After 1 h of incubation, excessive dye was removed by washing cells 2 times with 500 μL of phosphate buffered saline (PBS). Live cells were examined with a Leica TCS SP5 Scanning confocal microscope (Centre for Molecular and Cell Technologies at the SPbU) using Nomarski interference contrast (NIC) and confocal laser scanning microscopy (CLSM) with a 405nm laser.

### 3.6. Cell Culture Cultivation and Cytotoxicity Studies

To assess the cytotoxicity, 2 distinct cell lines were investigated, namely HEK293 and HeLa, due to their different properties and origins. The proportion of viable cells after the exposure to the compounds was determined using the MTT assay [42] by assessing their metabolic activity in the cell culture. HEK293 and HeLa cell cultures were grown in DMEM standard medium supplemented with 10% fetal bovine serum (FBS) at 37 °C in an atmosphere containing 5% CO_2_. The cells were transferred to 96-well plates (5000 cells per well in 100 μL DMEM + 10% FBS). The plates were incubated for 24 h, and culture medium was replaced with 100 μL DMEM + 10% FBS containing various concentrations of the examined compounds (5, 25, 50, 75, and 100 μM). After 24 h of incubation, 20 μL of MTT solution (3-(4,5-dimethylthiazol-2-yl)-2,5-diphenyltetrazolium bromide at a concentration of 5 mg/mL) was added to the wells. After 3 h of incubation, the medium was removed and 100 μL of DMSO was added to each well. Using a BioRad xMark microplate spectrophotometer, the absorbance of the resulting solutions was measured at 570 nm. The obtained values are directly proportional to the number of surviving cells after cultivation in the presence of the examined compounds. The percentage of cell viability in the presence of the examined compounds relative to non-treated cells was calculated.

## 4. Conclusions

A novel chromophore framework—4,5-bis(arylethynyl)-1*H*-1,2,3-triazole was developed. The general synthetic approach towards 4,5-bis(arylethynyl)-1*H*-1,2,3-triazole is based on Cu-catalyzed azide–alkyne cycloaddition of 1-iodobuta-1,3-diynes with organic azides and the Sonogashira coupling of resulting 5-iodo-4-ethynyl-1,2,3-triazoles with terminal alkynes. The approach allows for the varying of the nature and the position of the substituents in both aryl rings.

The promising photophysical properties for the discovered 4,5-bis(arylethynyl)-1*H*-1,2,3-triazoles were demonstrated. Thus, 4,5-bis(arylethynyl)triazoles differ from other known triazole dyes in their impressively high Stokes shifts (up to ~ 17.000 cm^−1^) and high fluorescence quantum yields (QY) (up to 60%). The QY and Stokes shift values depend on the position of EWG and EDG in both aryl rings. The most promising properties were found in the case of triazoles bearing *para*-EWG in the aryl ring at the C4 position along with *ortho*-EDG in the aryl ring at the C5 position of the 4,5-bis(arylethynyl)-1*H*-1,2,3-triazole core. Compared to 5-aryl-4-arylethynyltriazoles, the expansion of the conjugated system due to the addition of a C-C bond at the C5 position of 4,5-bis(arylethynyl)-1,2,3-triazoles makes it possible to shift the excitation wavelength towards the red region of the spectrum at 30–50 nm and to obtain more examples of fluorescent triazoles for further selection of the optimal compounds.

We demonstrated that the diethynyltriazole fluorescent dyes with an isothiocyanate group and with an azido group can be used to label native proteins and alkyne-functionalized proteins, respectively. The triazole dyes with the ester group are able to penetrate into cells, imparting a blue glow to the cells. Moreover, the developed triazole dyes were found to be nontoxic for various cell lines (HeLa and HEK293), which is very important for their further biological application.

## Data Availability

The presented data are available in this article.

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
