# Peer review of "4,5-Bis(arylethynyl)-1,2,3-triazoles—A New Class of Fluorescent Labels: Synthesis and Applications"

_molecules, 2022, doi:10.3390/molecules27103191_

Round 1
Author Response
Dear Reviewer,
Thank you very much for the interest to our manuscript, for careful reading, and for valuable comments and questions. We have addressed all of them.
- The way of writing seems to be naive. There are numerous grammatical and scientific errors throughout the manuscript. Therefore, the language must be improved.
We tried our best to improve the language
- 1st person narration is followed for synthesis.
There are two common styles: 1st person narration and passive voice. We describe all synthetic procedures in the passive.
- All the data obtained via spectroscopy and presented in paper must be in high resolution with minimum 300 dpi and the captions must be elaborative to make representation better.
Corrected
- It would be good if the authors would shed some light on the criteria for protein selection as far as protein labelling is concerned and the selection of HEK293 and HeLa cell lines for toxicity studies.
Hela and Hel293 are the most commonly used cancer cell lines for various biological studies. Therefore, we decided to use them to test membrane permeability with fluorescent triazoles.
- The authors have attributed less efficient labelling of 9a with ovalbumin, carbonic anhydrase and recombinant protein KNOX3-HD to different properties of proteins used in the standard mixture, e.g. amount of the lysine residues available for labelling. It would be more understandable if some more insights could be provided.
We rewrote this sentence. The corrected sentence has been added to the manuscript.
Isothiocyanate groups interact not only with the amino terminus of the protein but also with the side chain of lysine residues on the surface of the protein. The observed selectivity of labeling might be due to the different amounts of lysine residues on the surface of the proteins in the standard mixture.
- Some recent articles can be cited which have been left out:
1. Metals as “Click” catalysts for alkyne-azide cycloaddition reactions: An overview. DOI: https://doi.org/10.1016/j.jorganchem.2021.121846
2. Robust and Versatile Cu(I) metal frameworks as potential catalysts for azide-alkyne cycloaddition reactions: Review. DOI: https://doi.org/10.1016/j.mcat.2021.111432
Thank you very much for noticing those missing references. We add them to the manuscript.
Reviewer 2 Report
The current work of Balova and coworkers describes the synthesis of 4,5-bis(arylethynyl)-1H-1,2,3-triazoles and their applications in the fluorescent labeling of proteins. Previously the same group reported the design and synthesis of 5-aryl-4-Arylethynyl-1H-1,2,3-triazoles as fluorescent labels (Molecules 2021, 26, 2801 & J. Org. Chem. 2019, 84, 1925–1940). In continuation of their previous work, authors reported the 4,5-bis(arylethynyl)-1H-1,2,3-triazoles as a new class of fluorescent labels. The triazole derivatives have been synthesized in two steps, based on Cu-catalyzed azide-alkyne cycloaddition of 1-iodobuta-1,3-diynes with azides and the subsequent Sonogashira coupling of 5-iodo-4-ethynyl-1,2,3-triazoles with terminal alkynes. A detailed study of photophysical properties indicates that these derivatives show high stokes shifts and high fluorescence quantum yields (QY) (up to 60%). Importantly, these properties can be tuned based on EWG and EDG substitutions in both aryl rings. The triazoles containing the EWG functionality at the C4 position and ortho-EDG functionality at the C5 aryl ring displayed promising properties. Authors have also demonstrated the use of 4,5-bis(ar-ylethynyl)-1,2,3-triazoles in labeling of native proteins and azide-modified proteins. Importantly, these triazole fluorophores were found to be nontoxic toward HEK293 and HeLa cell lines. However, the use of UV light for the excitation could potentially limit the wide applicability of these fluorophores in biological applications. Despite the earlier work, I found that the previous reports do not endanger the current work’s novelty, and I suggest the manuscript can be accepted after major revision.
Comments:
- The authors claim that the “design of fluorophores and luminescent compounds based on triazole is a rather rare” a sentence that is at least questionable; probably, it should be better to say that a few examples have been reported. Moreover, the authors cited a review article for the previous examples, which clearly indicates that an extensive study was carried out in the field. A few important examples for the design of fluorophores based on the triazoles should be included in the manuscript.
- In the substate scope, incorporating an azetidine ring could be interesting. For examples, ethyl 2-(5-((4-(azetidin-1-yl)phenyl)ethynyl)-4-((4-cyanophenyl)ethynyl)-1H-1,2,3-triazol-1-yl)acetate. It is known that the azetidine containing TAMRA derivatives display improved photophysical properties.
- Authors should discuss the uniqueness and advantages of current work as compared to their previous work Ref 28 Molecules 2021, 26, 2801.
- On page 2, “We decided to extend this approach by changing the type of the last synthetic step, i.e., replacing the Suzuki with the Sonogashira cross-coupling” This sentence needs to be modified. In fact, both the synthetic routes have been disclosed in the previous JOC paper ( Org. Chem. 2019, 84, 1925–1940).
- On page 3, “For the synthesis of the target 1,2,3-triazoles 5 functionalized with an electron-donating (EDG, D) or an electron-with-drawing group (EWG, A) in C-4 and C-5 positions”. This sentence is misleading; most substrates consist of the EWG group at C-4 and EDG at the C-5 position.
- Scheme 1 requires a clear Aryl (Ar) substitution presentation. The scheme contains the “Ar” group, but the footnote depicts only the “R” group.
- On page 5, “On the other hand, the position of the substituents in the phetnylethynyl groups (or-tho- or para-) is extremely important electron donating dimethylamino group in the ortho-position of the arylethynyl fragment reveals the stronger bathochromic shift (447 vs 429 nm) and a larger Stokes shift (192 vs 137 nm) compared to the values observed for triazole 5d with para-NMe2-group in the phenylethynyl fragment. The photophysical properties of 5a/5c are different (opposite) from the 5b/5f and 5e/5h. Can the authors explain this anomaly in the properties with 5a/5c.
- Authors should pay a closer look into the numbering of compounds in the manuscript. For example, On page 9, “Azide 8c is almost non-fluorescent in water”. It should be 8b instead of 8c.
On page 10, “Bovine serum albumin (BSA) and aldolase were labeled separately (Fig. 8a).” The correct figure number is Fig. 7a
On page 10, “we used two binding ways: the reaction of azide-alkyne cycloaddition (for 8b)” the correct compound number is 8a.
- The manuscript requires closer attention to the English in the whole manuscript. For example; On page 2, “The choice of 2-az-idoethylacetate as the dipolarophile caused by the potential ability for further derivatization of the ester group”
On page 3, “If the emission intensity for 8a and 9a in water is almost the same”
On page 3, Figure 1 “photophysics propertes"
These sentences should be modified/corrected.
- On In page 12, “HeLa cells were cultured in media with fluorescent dyes (5d) for only an hour and then analyzed by confocal laser scanning microscopy (Fig.10). The excess dye was removed by two washing steps, and the authors should mention the washing steps in the main text. Authors should also include the negative control image in Figure 10.
Author Response
Dear Reviewer,
Thank you very much for the interest to our manuscript, for careful reading, and for valuable comments and questions. We have addressed all of them.
1. The authors claim that the “design of fluorophores and luminescent compounds based on triazole is a rather rare” a sentence that is at least questionable; probably, it should be better to say that a few examples have been reported. Moreover, the authors cited a review article for the previous examples, which clearly indicates that an extensive study was carried out in the field. A few important examples for the design of fluorophores based on the triazoles should be included in the manuscript.
We rewrote this part and added a few important examples of fluorophores based on the triazoles.
2. In the substrate scope, incorporating an azetidine ring could be interesting. For examples, ethyl 2-(5-((4-(azetidin-1-yl)phenyl)ethynyl)-4-((4-cyanophenyl)ethynyl)-1H-1,2,3-triazol-1-yl)acetate. It is known that the azetidine containing TAMRA derivatives display improved photophysical properties.
Thank you very much for a very interesting idea. The work is ongoing, so we will definitely try to synthesize such compounds.
3. Authors should discuss the uniqueness and advantages of current work as compared to their previous work Ref 28 Molecules 2021, 26, 2801.
It was done. The discussion has been added to the Introduction and Results and Discussion.
Recently, we have reported the synthesis of 5-aryl-4-arylethynyltriazoles with the 1,2,3-triazole core as a π-spacer between EWG and EDG attached to the C4 and C5 positions. 5-Aryl-4-arylethynyltriazoles were found to be fluorescent compounds with high Stokes shift (>100 nm) and promising fluorescence quantum yield (15-64%). [23] In order to study whether the extension of the conjugated π-system would improve the fluorescent properties of the triazole-based fluorophores, we decided to replace the aryl ring at the C5 atom with the arylethynyl moiety.
Therefore, here we report the efficient synthetic rout towards 4,5-bis(arylethynyl)-1,2,3-triazole bearing EWG and EDG at ortho- and para-positions of aryl rings. We demonstrated that easily synthetically accessible 4,5-bis(arylethynyl)-1,2,3-triazoles are fluorescent in a wider spectral range (350 – 600 nm. Moreover, some derivatives of new 4,5-bis(arylethynyl)-1,2,3-triazoles have extremely high Stockes shift values (up to 230 nm). The photophysical properties of 4,5-bis(arylethynyl)-1,2,3-triazoles are strongly dependent on the relative orientation of the EWG and EDG at the C4 and C5 position and on the type of the substitution (either ortho- or para-). To reach the highest fluorescence QY it is important to have EDG-ortho-substituted arylethynyl ring at the C5 position along with EWG-para-substituted arylethynyl ring at the C4 atom.
For the highest redshift of emission, the orientation of the groups must be inverted.
It should be noted, that the expansion of the conjugated system does not introduce fundamental changes in the photophisical properties of 4,5-bis(arylethynyl)-1,2,3-triazoles compared to 5-aryl-4-arylethynyltriazoles. However, it allows shifting the excitation wavelength to the red region of the spectrum by 30–50 nm and getting more examples of fluorescent triazoles for further selection of the optimal compounds.
4. On page 2, “We decided to extend this approach by changing the type of the last synthetic step, i.e., replacing the Suzuki with the Sonogashira cross-coupling” This sentence needs to be modified. In fact, both the synthetic routes have been disclosed in the previous JOC paper ( Org. Chem. 2019, 84, 1925–1940).
We changed this part focusing mainly on 4,5-bis(arylethynyl)triazoles as the object for the search of new fluorescent triazoles with extended π-system and improved photophisical properties.
5. On page 3, “For the synthesis of the target 1,2,3-triazoles 5 functionalized with an electron-donating (EDG, D) or an electron-with-drawing group (EWG, A) in C-4 and C-5 positions”. This sentence is misleading; most substrates consist of the EWG group at C-4 and EDG at the C-5 position.
We corrected this sentence to make it clearer.
The target 4,5-bis(arylethynyl)-1,2,3-triazoles 5 with electron-donating and electron-withdrawing groups in arylathynyl moieties at the C-4 and C-5 atoms of the triazole ring, were obtained using CuAAC of iodoaryldiacetylenes 1a‒d with 2-azidoethylacetate 2 followed by the Sonogashira coupling of 4-ethynyl-5-iodo-1,2,3-triazoles 3 with arylacetylenes 4a‒f with Pd(PPh3)4/K3PO4 as a catalytic system (Scheme 1).
6. Scheme 1 requires a clear Aryl (Ar) substitution presentation. The scheme contains the “Ar” group, but the footnote depicts only the “R” group.
Corrected
7. On page 5, “On the other hand, the position of the substituents in the phetnylethynyl groups (or-tho- or para-) is extremely important electron donating dimethylamino group in the ortho-position of the arylethynyl fragment reveals the stronger bathochromic shift (447 vs 429 nm) and a larger Stokes shift (192 vs 137 nm) compared to the values observed for triazole 5d with para-NMe2-group in the phenylethynyl fragment. The photophysical properties of 5a/5c are different (opposite) from the 5b/5f and 5e/5h. Can the authors explain this anomaly in the properties with 5a/5c.
The corrected discussion has been added to the manuscript.
On the other hand, the position of the substituents in the phetnylethynyl groups (ortho- or para-) is extremely important. Comparing pairs of triazoles with the similar substituents which differ only in the ortho/para position of the substituents, i.e. pair 1 (С4: para CN and С5: ortho-NMe2 (5a) / para-NMe2 (5c)); pair 2 (С4: para Cl and С5: ortho-NMe2 (5b) / para-NMe2 (5d)); pair 3 (С4: para Cl and С5: ortho-OMe (5f) / para-OMe (5g)) it is obvious, that QYs are always higher for the ortho-derivatives. Moreover, the Stockes shifts values for all pairs also have the similar trend: the values are smaller for para-isomers.
As for the mechanism of fluorescence. Several fluorescence mechanisms can be suggested for these complex systems: ICT, TICT, PLTICT, since the triazole ring can serve both as the EDG/EWG itself and as a pi linker between the EWG and EDG at positions C4/C5. In addition, a competitive PET mechanism can be proposed for the excited state decay. It is not possible to assign an exact mechanism to each molecule without DFT calculations. This requires an additional large theoretical study, which is planned.
8. Authors should pay a closer look into the numbering of compounds in the manuscript. For example, On page 9, “Azide 8c is almost non-fluorescent in water”. It should be 8b instead of 8c.
On page 10, “Bovine serum albumin (BSA) and aldolase were labeled separately (Fig. 8a).” The correct figure number is Fig. 7a
Corrected
On page 10, “we used two binding ways: the reaction of azide-alkyne cycloaddition (for 8b)” the correct compound number is 8a.
Corrected
9. The manuscript requires closer attention to the English in the whole manuscript. For example; On page 2, “The choice of 2-az-idoethylacetate as the dipolarophile caused by the potential ability for further derivatization of the ester group”
On page 3, “If the emission intensity for 8a and 9a in water is almost the same”
Corrected
On page 3, Figure 1 “photophysics propertes"
Corrected
10. On In page 12, “HeLa cells were cultured in media with fluorescent dyes (5d) for only an hour and then analyzed by confocal laser scanning microscopy (Fig.10). The excess dye was removed by two washing steps, and the authors should mention the washing steps in the main text. Authors should also include the negative control image in Figure 10.
Corrected
Reviewer 3 Report
The manuscript “4,5-Bis(arylethynyl)-1,2,3-triazoles – a new class of fluorescent labels: Synthesis and Applications” by Anastasia Govdi et al. reports on the synthesis of a series of triazoles by Cu-catalyzed cycloaddition followed by Sonogashira cross-coupling. The end products were investigated for fluorescence properties and used for protein labeling. In general, the manuscript comprises the three parts (synthesis, characterization and properties) that makes the study complete. The design, selection and execution of the “experiments” are appropriate and carried out correctly. I have only major one concern reading throughout the manuscript e.g. the English language. The manuscript needs to be rechecked very carefully in that regard. Thus the recommendation is for minor language corrections.
Can the HRMS - ESI spectra be added in the supporting information? The assessment of the purity of the compounds solely based on the NMRs is quite tricky. Probably one X-ray table for most important parameters ( as generated by Olex2) will be a plus ( also SI).
Please recheck that normalized intensity for the Fluorescence – in the figures. In Figure 3 (or in the text) the combination λ ex/em should be provided for all compounds. Besides, the discussion of fluorescence intensity is not helpful if the intensity is normalized. Actually, how did you perform the “normalization” as in the different figures the intensity is always “normalized” but e.g. for fig.5e and 5i almost zero while in fig. 1 is always up to one?
Minor
Please provide the Stockes shifts in nm.
The English language and meaning of the sentence needs to be corrected. As it is, the sentence is incomprehensible: „The nature of the objects under study is of crucial importance, because it required special photophysical properties and the possibility to introduce the functional groups reactive in different bioconjugative transformations bind a biotarget.
Ibid for the sentence (too long please consider splitting etc.):
“The design of fluorophores with location of the donor and acceptor parts at 4-th and 5-th positions of the triazole ring can avoid interruption of conjugation between them through N1 nitrogen atoms of the triazole ring, and although this does not lead to a significant shifts in the absorption maximum, such compounds exhibit the most intense absorption [25].”
The cytotoxicity is usually assed not on “cancer” cell lines e.g. HeLa and HEK293? Mean, those are also cancer cell lines but … don’t know how to say it exactly.
Introduction: “In particular, fluorescent imaging and flow cytometry should considered as …” Suggestion: …. should be considered …
Next sentence: “One of the important steps in, these investigations is a probe selection with the appropriate properties, that allow for the … “
Suggestion: definitely “allows” and probably “… is a probe selection with appropriate properties…” Please check.
…selectivity labeling or selective labelling. Though I think to understand the use of“ small molecule size” it is a jargon. Could you please try to rephrase? If not synthetic, but natural, a small molecule size fluorescent probe is disadvantageous?
“That limit” or limits?
“In addition, the dyes must meet such requirements as ease of synthesis …” The authors should make a distinction between a dye and fluorescent probe molecules. The use of a dye is not the same as a fluorescent probe. Please correct.
Figure 1: In the previous works there are some C4-N1 linkers . The different types (other than C4-N1) are not referred. Is it a drawing mistake or …. ?
“However, the fluorescence intensity of these compounds is much lower than for other triazoles 5.” Please specify ‘these” compounds. Do you mean 5h and 5i ?
In table 1 The column label after ΦF , % b is not readable. Are the values λ ex/em for PBS or THF ?
“ …..for subsequent modification this compounds with functional groups”. Suggestion: …modification of this compounds…
“crosslinking Spaser” to crosslinking spacer
PBS buffer and DMEM are also water solutions. Please consider changing aqueous solutions to water. “The fluorescence intensity for most compounds in PBS buffer and DMEM increases in comparison with aqueous solutions (Fig.6).”
Would suggest to rearrange figure 6 “by compound” instead of “by solvent”. E.g. put 9a in PBS water DMEM ABS /Em on one and the same plot/ “picture”.
“….allow using an azide-alkyne cycloaddition to introduce fluorescent labels into metabolically modified proteins or other biomolecules”. Not sure about the term metabolical in respect to proteins (wild type vs AA mutation, …) or did the authors intend post-translational modifications (glycosylation, acetylation etc.). Besides the sentence is confusing and needs to be rearranged .
P9 “If the emission intensity for 8a and 9a in water is almost the same” Please remove the “ If “ – if correctly understood.
Table 1 and Table2 may be merged .
“It has been shown that the binding of triazolylisothiocyanate 9a is most effective with proteins, which mass is greater than 66 kDa.” The aldolase emission 40kDa is quite intense. Probably the AA sequence plays a role. Why not try also Lysozyme ~13kDa, proteinase K ~30kDa, DNA ligase above 80kDa … those are common in labs (not mandatory just a suggestion).
The sentence below fig. 9 is interesting. If “ a lot of non-specific, noncovalent hydrophobic bindings of BSA with triazole dye 8a” is also detected why do you need the covalent binding ? Can you use “directly” ?
3.1. General Information “…for F-containing compounds….”?? Please precise
3.2.3. First sentence THF (0.05 M) … aqueous solution or LiOH ~0.05M or other solution ….. Please clarify
3.3 ….equipped using an integrating sphere …suggestion equipped with an integrating sphere. Any chance to say the size of the sphere (diameter)
In the conclusions it is true that proteins have been modified but would suggest labeling as the function has not been altered.
That’s all .
Author Response
Dear Reviewer,
Thank you very much for the interest to our manuscript, for careful reading, and for valuable comments and questions. We have addressed all of them.
Please provide the Stockes shifts in nm.
Corrected
The English language and meaning of the sentence needs to be corrected. As it is, the sentence is incomprehensible: „The nature of the objects under study is of crucial importance, because it required special photophysical properties and the possibility to introduce the functional groups reactive in different bioconjugative transformations bind a biotarget.
The nature of the objects under study is of crucial importance, because it requires special photophysical properties and the ability to vectorize the label core by introducing the functional groups reactive in different bioconjugative transformations to bind a biotarget [9].
Ibid for the sentence (too long please consider splitting etc.):
“The design of fluorophores with location of the donor and acceptor parts at 4-th and 5-th positions of the triazole ring can avoid interruption of conjugation between them through N1 nitrogen atoms of the triazole ring, and although this does not lead to a significant shifts in the absorption maximum, such compounds exhibit the most intense absorption [25].”
This sentence has been deleted. The introduction part has been changed.
The cytotoxicity is usually assed not on “cancer” cell lines e.g. HeLa and HEK293? Mean, those are also cancer cell lines but … don’t know how to say it exactly.
HeLa and HEK293 cell lines are widely used for cytotoxicity studies of a wide variety of compounds, ranging from plant extracts to nanoparticles. The use of these lines also allow comparison between different studies.
Appropriate explanations have been added to the text of the article.
Introduction: “In particular, fluorescent imaging and flow cytometry should considered as …” Suggestion: …. should be considered …
Responding to this and all subsequent the Reviewer’s remarks by regarding the introduction. They inspired us and the introduction part has been completely rewritten.
Next sentence: “One of the important steps in, these investigations is a probe selection with the appropriate properties, that allow for the … “
Suggestion: definitely “allows” and probably “… is a probe selection with appropriate properties…” Please check.
…selectivity labeling or selective labelling. Though I think to understand the use of“ small molecule size” it is a jargon. Could you please try to rephrase? If not synthetic, but natural, a small molecule size fluorescent probe is disadvantageous?
If not synthetic, but natural, a small molecule size fluorescent probe is disadvantageous?
“That limit” or limits?
“In addition, the dyes must meet such requirements as ease of synthesis …” The authors should make a distinction between a dye and fluorescent probe molecules. The use of a dye is not the same as a fluorescent probe. Please correct.
Done
Figure 1: In the previous works there are some C4-N1 linkers . The different types (other than C4-N1) are not referred. Is it a drawing mistake or …. ?
The drawing has been corrected and a discussion has been added
“However, the fluorescence intensity of these compounds is much lower than for other triazoles 5.” Please specify ‘these” compounds. Do you mean 5h and 5i ?
Yes, 5h and 5i numbers were added
In table 1 The column label after ΦF , % b is not readable. Are the values λ ex/em for PBS or THF ?
The values are given for THF solutions. To make it clearer, the solvent type was added to the column name.
“ …..for subsequent modification this compounds with functional groups”. Suggestion: …modification of this compounds…
Corrected
With different fluorescent triazoles in hand, we tuned to converting the ester group into various functional groups suitable for further conjugation with biomolecules. Two triazoles 5b,h with high fluorescence intensity, suitable emission wavelength (λem > 440 nm) and large Stokes shift were selected.
“crosslinking Spaser” to crosslinking spacer
Corrected
PBS buffer and DMEM are also water solutions. Please consider changing aqueous solutions to water. “The fluorescence intensity for most compounds in PBS buffer and DMEM increases in comparison with aqueous solutions (Fig.6).”
Corrected
The absorption and fluorescence spectra of triazoles in aqueous media depend on the nature of the aqueous solution (Fig. 5). The fluorescence for the solutions of most compounds in PBS buffer and DMEM is more intense in comparison with those observed for the solutions in water (Fig. 5).
Would suggest to rearrange figure 6 “by compound” instead of “by solvent”. E.g. put 9a in PBS water DMEM ABS /Em on one and the same plot/ “picture”.
Corrected
“….allow using an azide-alkyne cycloaddition to introduce fluorescent labels into metabolically modified proteins or other biomolecules”. Not sure about the term metabolical in respect to proteins (wild type vs AA mutation, …) or did the authors intend post-translational modifications (glycosylation, acetylation etc.). Besides the sentence is confusing and needs to be rearranged .
Corrected
In addition, the functionalization of triazoles with an azido group allows using an azide-alkyne cycloaddition to introduce triazole-based fluorescent labels into biomolecules premodified by alkyne-type functional groups.
P9 “If the emission intensity for 8a and 9a in water is almost the same” Please remove the “ If “ – if correctly understood.
Corrected
Table 1 and Table2 may be merged .
Thank you very much for this suggestion. However, it is better to leave these parts unchanged. Thus, the information in Table 1 and the related discussion refers to the unmodified fluorescent triazole. On the other hand, Table 2 and the related discussion is for triazoles modified with additional conjugated groups. Combining the information of both tables would complicate understanding the discussed parts.
“It has been shown that the binding of triazolylisothiocyanate 9a is most effective with proteins, which mass is greater than 66 kDa.” The aldolase emission 40kDa is quite intense. Probably the AA sequence plays a role. Why not try also Lysozyme ~13kDa, proteinase K ~30kDa, DNA ligase above 80kDa … those are common in labs (not mandatory just a suggestion).
Thank you very much for the suggestion. We will definitely try in in the future research.
The sentence below fig. 9 is interesting. If “ a lot of non-specific, noncovalent hydrophobic bindings of BSA with triazole dye 8a” is also detected why do you need the covalent binding ? Can you use “directly” ?
Thank you for this suggestion. Indeed, we found the noncovalent type of binding only for the covalently bound probes. However, the research devoted directly to the possibility of application of noncovalent type of binding looks very promising. We will study it in future. The corresponding part has been corrected.
The BSA was successfully labeled with NSC-triazoles 8a. However, to our surprise, many non-specific, non-covalent hydrophobic types of BSA binding to triazole dye 8a were found. Thus, in the UV-stained electrophoregram, along with labeled BSA, the presence of free dye molecules was observed. This finding opens the way for further possible applications of non-covalent lableling of proteins with triazole-based fluorescent dyes.
3.1. General Information “…for F-containing compounds….”?? Please precise
This sentence has been deleted.
3.2.3. First sentence THF (0.05 M) … aqueous solution or LiOH ~0.05M or other solution ….. Please clarify
Corrected
To a stirred solution of triazoles 5 (1 equiv) in THF (15 mL) was added aqueous solution of LiOH (2-4 equiv, 0.05 M)
3.3 ….equipped using an integrating sphere …suggestion equipped with an integrating sphere. Any chance to say the size of the sphere (diameter)
Corrected
Equipped with an integrating sphere
In the conclusions it is true that proteins have been modified but would suggest labeling as the function has not been altered.
Thank you for the comment. We have replaced «modify» with «label»
We demonstrated that the diethyneyltriazole fluorescent dyes with an isothiocyanate group and with an azido group can be used to label native proteins and alkyne-functionalized proteins, respectively.
Round 2
Reviewer 2 Report
The authors have revised the original version of the manuscript by appropriately answering the questions and comments raised by reviewers. However, I was a bit confused with Figure 10.
Figure 10 is labeled as control (b) and cells with probe 5d as (a). However, the image indicates the otherwise. (a) control cells, (b) cells were incubated for 1 h. The authors should examine the manuscript carefully.
I recommend this manuscript for publication.